# Developing a QSPR Model of Organic Carbon Normalized Sorption Coefficients of Perfluorinated and Polyfluoroalkyl Substances

**DOI:** 10.3390/molecules27175610

**Published:** 2022-08-31

**Authors:** Lan Jiang, Yue Xu, Xiaoyu Zhang, Bingfeng Xu, Ximeng Xu, Yixing Ma

**Affiliations:** 1Faculty of Civil Engineering and Mechanics, Kunming University of Science and Technology, Kunming 650500, China; 2Yunnan Research Academy of Eco-environmental Sciences, Kunming 650034, China; 3Faculty of Environmental Science and Engineering, Kunming University of Science and Technology, Kunming 650500, China

**Keywords:** perfluorinated and polyfluoroalkyl substances, organic carbon normalized sorption coefficient, quantitative structure–property relationship, distribution behavior

## Abstract

Perfluorinated and polyfluoroalkyl substances (PFASs) are known for their long-distance migration, bioaccumulation, and toxicity. The transport of PFASs in the environment has been a source of increasing concerned. The organic carbon normalized sorption coefficient (K_oc_) is an important parameter from which to understand the distribution behavior of organic matter between solid and liquid phases. Currently, the theoretical prediction research on log K_oc_ of PFASs is extremely limited. The existing models have limitations such as restricted application fields and unsatisfactory prediction results for some substances. In this study, a quantitative structure–property relationship (QSPR) model was established to predict the log K_oc_ of PFASs, and the potential mechanism affecting the distribution of PFASs between two phases from the perspective of molecular structure was analyzed. The developed model had sufficient goodness of fit and robustness, satisfying the model application requirements. The molecular weight (*MW*) related to the hydrophobicity of the compound; lowest unoccupied molecular orbital energy (*E*_LUMO_) and maximum average local ionization energy on the molecular surface (*ALIE*_max_), both related to electrostatic properties; and the dipole moment (*μ*), related to the polarity of the compound; are the key structural variables that affect the distribution behavior of PFASs. This study carried out a standardized modeling process, and the model dataset covered a comprehensive variety of PFASs. The model can be used to predict the log K_oc_ of conventional and emerging PFASs effectively, filling the data gap of the log K_oc_ of uncommon PFASs. The explanation of the mechanism of the model has proven to be of great value for understanding the distribution behavior and migration trends of PFASs between sediment/soil and water, and for estimating the potential environmental risks generated by PFASs.

## 1. Introduction

Perfluorinated and polyfluoroalkyl substances (PFASs), as a class of synthetic aliphatic compounds [1,2], have be widely used in industrial production and daily consumer products because of their hydrophobicity, oleophobicity, thermal stability, and chemical stability [3,4]. Up to now, PFASs and their precursors have been found in water, atmosphere, soil, sediment, and other environmental media [5,6,7,8]. Generally, the concentration of PFAS in these environmental media is within the scale range of ng·L^−1^, but in some areas with serious pollution (i.e., around fluorine chemical plants), the concentration of PFAS in water can reach the scale range of mg·L^−1^ [9]. PFASs can enter the human body and accumulate through drinking water, the food chain, and in other ways [10,11], and when a certain threshold is reached, they will produce corresponding toxic effects, such as neurological, reproductive, liver, and endocrine toxicity, which can seriously endanger human health [12,13,14,15,16].

Sediment, soil, and water are important sinks for PFASs [1,10,17]. The accurate measurement of the organic carbon normalized sorption coefficient (K_oc_) of PFASs can reflect their distribution behavior between sediment/soil and water [18,19,20], which is crucial for their environmental fate and risk assessment. So far, many studies have been undertaken on the distribution behavior of traditional PFASs, such as perfluorinated carboxylic acids (PFCAs) and perfluoroalkyl sulfonic acids (PFSAs), but there are few studies on that of emerging PFASs. PFASs are composed of a carbon skeleton and hydrophilic groups, where the hydrogen atoms connected to the carbon skeleton are partially or completely replaced by fluorine atoms [1,2]. From a structural perspective, both conventional and emerging PFASs are dominated by the carbon skeleton. However, the substituents on the skeleton and the functional groups at the ends have an important impact on the environmental transport characteristics of the compounds. Previous studies have shown that the migration of PFASs in different environmental media is closely related to their structural factors, such as their carbon chain length, substituents, and functional groups [17,21]. For instance, for the two-phase medium of sediment/soil and water, there is a linear relationship between the K_oc_ of PFASs and the number of perfluorinated carbon (CF). In general, the K_oc_ increases with the number of CF, while PFASs with a sulfonic group have a larger K_oc_ than similar compounds containing a carboxyl group [21]. The lack of research on the environmental migration law of emerging PFASs results from the variety of PFASs, the fact that their derivatives appeared one after another, and the insufficient understanding of their physicochemical properties. At the same time, the experimental measurement of K_oc_ is not only cumbersome and costly but may also pose environmental pollution and human health risks in large-scale experiments. However, it is possible to quickly fill the data gap syrrounding the log K_oc_ of PFASs at low experimental cost by constructing a mathematical model based on the structural characteristics of PFASs to predict their K_oc_.

The quantitative structure–property relationship (QSPR) model is a theoretical prediction tool with a rapid development and a wide application range. It establishes a functional relationship between the molecular structure of compounds and their properties to effectively predict the compounds’ properties [22,23]. The QSPR model can be used to predict the partition coefficient of various organic pollutants with high efficiency, such as the partition coefficient of polycyclic aromatic hydrocarbons (PAHs) between polydimethylsiloxane (PDMS) and water [24], the partition coefficient of polychlorinated biphenyls (PCBs) between low-density polyethylene and water [25], and the partition coefficient of PFASs between gas and particles [2]. To date, few studies have employed PFASs as a unique research object to construct QSPR models for predicting their log K_oc_ [26,27]. A previous study reported the log K_oc_ of 824 organic compounds predicted by a QSPR model, but only a few PFASs were included [26]. Due to the limitation of its data set, the application scope of the model was narrow for PFASs Another limitation of prior work on the model’s establishment was that the modeling process did not fully follow the five guidelines for QSPR model construction [27,28]. Generally, for the log K_oc_ prediction of PFASs by QSPR, it is necessary to improve the applicability and accuracy of the model. Meanwhile, the standardization of modeling also need further investigation. Owing to the issues with the above models, there is still a knowledge gap in the overall analysis of PFASs distribution mechanism between sediment/soil and water at the molecular level.

This study developed an optimal log K_oc_ prediction model for PFASs based on the QSPR model construction guidelines. A comprehensive verification and evaluation of the model was undertaken to ensure the integrity and standardization of the modeling process and achieve a reasonable prediction of the log K_oc_ of PFASs. The model used 22 PFASs from eight different classes as the modeling dataset, equipped with strong pertinence which greatly improves the applicability of the model for PFASs. Molecular descriptors with clear definitions were included in the model, which identified the potential mechanism of PFASs distribution between sediment/soil and water from the perspective of the molecular level quickly and accurately, facilitating a better understanding of the distribution behavior of PFASs between the two phases. This combination of effects has significant practical implications for enriching the migration theory of PFASs with different structures between sediment/soil and water, and provides a reference for predicting the deposition concentration of emerging PFASs in environmental media.

## 2. Results and Discussion

### 2.1. Model Construction and Validation

After stepwise linear regression, the optimal QSPR model (Equation (1)) was obtained. The model contained four molecular descriptors: molecular weight (*MW*), dipole moment (*μ*), lowest unoccupied molecular orbital energy (*E*_LUMO_), and maximum average local ionization energy on the molecular surface (*ALIE*_max_).
log K_oc_ = 7.334 × 10^−3^ *MW* − 1.705 *μ* − 0.956 *E*_LUMO_ − 1.398 *ALIE*_max_ + 24.10(1)

Appendix A lists the calculated values of the four molecular descriptors and the predicted values of log K_oc_. The statistical parameters of the developed model are shown in Table 1. According to the evaluation criteria of the QSPR model summarized in previous studies [29], coefficient of determination (*R*^2^) > 0.8, multiple correlation coefficient of leave-one-out cross-validation (*Q*^2^_LOO_) > 0.5, external validation indicators (*Q*^2^_F1_, *Q*^2^_F2_, and *Q*^2^_F3_) > 0.5, indicating that the model has sufficient goodness of fit, robustness, and predictive ability and meets the requirements of the QSPR model construction guidelines [28]. In addition, the *R*^2^ − *Q*^2^_LOO_ value of this model is less than 0.3, indicating that there is no overfitting phenomenon in this model [30]. *Q*^2^_LOO_, *Q*^2^_F1_, *Q*^2^_F2_, and *Q*^2^_F3_ were calculated according to the method in a previous study [29], and the calculation formulas of these parameters are presented in the Appendix A; *R*^2^ was obtained using SPSS 26 software (IBM SPSS Inc., Chicago, IL, USA).

Figure 1 shows the error distribution of the model prediction. The prediction errors of the PFASs were randomly distributed on both sides of the baseline, and there was no obvious regularity, indicating that the built model had no obvious systematic errors. Table 2 lists the significance index (*p*) and variance inflation coefficient (*VIF*) values of the molecular descriptors contained in the model. When *p* < 0.05, this indicates that the molecular descriptor was significant; when *VIF* < 10, this indicates that there was no multicollinearity among the molecular descriptors [31]. It can be seen from the table that all molecular descriptors in the model were key descriptors, and there was no collinearity among them. *p* and *VIF* were obtained using SPSS 26 software (IBM SPSS Inc., Chicago, IL, USA).

As shown in Figure 2, the good consistency between the predicted value and the experimental value indicates that the established model has high prediction accuracy for the log K_oc_ of the PFASs.

### 2.2. Application Domain

According to the Williams plot (Figure 3), none of the standardized residuals of the log K_oc_ of the PFASs obtained from the QSPR model exceeded the thresholds (−3, 3) [1]. In the test set, leverage values (*h*) > warning leverage value (*h**) (*h** = 0.83) of 6:2 chlorinated polyfluorinated ether sulfonate (6:2 Cl-PFAES), indicating that this substance was structurally quite different from most of the PFASs in the training set; 6:2 Cl-PFAES is an emerging PFAS that has been widely used in industry as a substitute for traditional PFASs (such as perfluorooctane sulfonic acid (PFOS)) [32,33]. The QSPR model predicts that the standardized residual of its log K_oc_ value is 0.454, which does not exceed the threshold. It can be seen that the QSPR model has a wide range of applications and strong generalization ability, which can successfully predict not only traditional PFASs but also emerging PFASs.

### 2.3. Mechanistic Interpretation of the Model

The standardized regression coefficient refers to the regression coefficient when all variables are expressed in standardized form. Since the same measurement unit is used, the independent variables are more comparable [34]. The standardized regression coefficients of *MW*, *μ*, *E*_LUMO_, and *ALIE*_max_ in the QSPR model were 1.048, −0.219, −0.495, and −0.362, respectively. Based on a comparison of their absolute values, it can be seen that the influence of the four molecular descriptors on log K_oc_ was *MW* > *E*_LUMO_ > *ALIE*_max_ > *μ*.

*MW* is related to the molecular size and hydrophobicity and can reflect the effect of molecules on the formation and destruction of holes in water [35]. When *MW* increases, this increases the PFASs’ molecular size and strengthens their hydrophobicity [36,37]. At this time, the energy of the adsorbate required for the formation of holes between water molecules will lead to stronger hydrophobic repulsion of water on the surface of the PFAS molecules [38], thus driving the adsorption of PFASs in solid media (such as sediment or soil) [17]. In this study, the QSPR model showed that *MW* was positively correlated with log K_oc_. When *MW* increased, the log K_oc_ of PFSAs showed an increasing trend, and the log K_oc_ of PFCAs roughly showed an increasing trend (except for perfluorobutanoic acid (PFBA); the possible explanation is given below). This result is consistent with the previously reported results that the log K_oc_ of the same type of PFASs usually increases with the increase in the carbon chain [39]. PFASs of the same class have similar structures (with the same functional groups). With the increase in the carbon chain length, its *MW* and hydrophobicity increased, which promoted the adsorption of PFASs in solid medium and increased their log K_oc_. In addition, the size of *MW* explained the change of log K_oc_ of different types of PFASs to a certain extent. For example, perfluorododecanoic acid (PFDoDA), 6:6 perfluoroalkyl phosphinic acid (6:6 PFPiA), and n-ethyl perfluorooctane sulfonamidoacetic acid (N-EtFOSAA) have the same carbon chain length (12 carbon atoms); their *MW* are 6:6 PFPiA > PFDoDA > N-EtFOSAA; their *μ* are N-EtFOSAA > PFDoDA > 6:6 PFPiA; their *E*_LUMO_ are PFDoDA > 6:6 PFPiA > N-EtFOSAA; their *ALIE*_max_ are N-EtFOSAA > 6:6 PFPiA > PFDoDA, and their log K_oc_ are 6:6 PFPiA > PFDoDA > N-EtFOSAA, which shows the same ranking as *MW*.

*E*_LUMO_ reflects the ability of molecules to receive electrons [40]. When *E*_LUMO_ is more positive, it is more difficult for molecules to obtain electrons from the external environment. *ALIE*_max_ is the average energy required to ionize electrons at any point in molecular space [41], which is related to their molecular electrostatic potential, electronegativity, hardness, and other properties [42]. When *ALIE*_max_ is smaller, the electron activity is stronger, and electrophilic and free radical reactions are more likely to occur [43]. According to the QSPR model, *E*_LUMO_ and *ALIE*_max_ were negatively correlated with log K_oc_, which reflects the electrostatic interaction between molecules. These two molecular descriptors were used to explain why the molecular weights of perfluoroheptanoic acid (PFPeA) and perfluorohexanoic acid (PFHxA) were both larger than that of PFBA but their log K_oc_ was smaller than that of PFBA. From the perspective of *E*_LUMO_, it is more difficult for PFBA to obtain electrons from the external environment than PFPeA and PFHxA, but the difference between the compounds is small. From the perspective of *ALIE*_max_, PFBA has stronger electronic activity than PFPeA and PFHxA and is more prone to electrophilic reactions, and the difference between the compounds is larger. Compared with PFPeA and PFHxA, PFBA may be more capable of electrostatic interaction with the external environment, increasing its log K_oc_. This explanation is consistent with the results reported previously that electrostatic interactions may be the main factor affecting the adsorption of short-chain PFASs in solid-phase media [44].

*μ* is often used to describe the intermolecular dipole–dipole interactions in QSPR studies [45]. As *μ* increases, ionic compounds are more easily solvated in liquid phases (water) [46], and PFASs are adsorbed with greater difficulty in solid phases (aqueous aerosol) [1]. It can be seen from the QSPR model that *μ* was negatively correlated with log K_oc_. The larger the *μ* value, the easier it is to solvate in water, so it is more difficult for PFASs to be adsorbed in solid media. For example, N-EtFOSAA is the precursor of perfluorooctane sulfonamide (PFOSA) [47]; the two have similar structures and they have the same carbon chain length; only the terminal functional groups are different. The *MW* of N-EtFOSAA is larger than that of PFOSA, and the *ALLE*_max_ is smaller, but the log K_oc_ of PFOSA is much larger than that of N-EtFOSAA. In addition to the influence of *E*_LUMO,_ it is mainly due to the *μ* of N-EtFOSAA being relatively large (nearly threefold).

### 2.4. Model Comparison

The QSPR model with high *R*^2^ was developed using the molar volume as a single molecule descriptor [27], but the robustness and external prediction ability of the model were not verified in the process of model construction, and the log K_oc_ prediction ability of PFASs with molar volume less than 160 cm^3^·mol^−1^ was limited. In this study, the robustness and external prediction ability of the developed QSPR model were verified through the *Q*^2^_LOO_ and *Q*^2^_F1_, *Q*^2^_F2_, and *Q*^2^_F3_. The results show that the model has good robustness and external prediction ability. For example, the model built by Brusseau had prediction errors of 1.4 and 0.5 for PFBA and PFPeA (two PFASs with molar volumes of less than 160 cm^3^·mol^−1^) [27], while the QSPR model developed in this study had prediction errors of 0.52 and −0.13 for PFBA and PFPeA, respectively. The log K_oc_ prediction ability of this model is better for PFASs with a smaller molar volume.

A QSPR model was developed using nine molecular descriptors based on the structural properties of 824 compounds [26], but only six PFASs were included in these compounds. The QSPR model established by Wang et al. has a lower prediction accuracy of log K_oc_ for PFASs than the QSPR model, which only takes PFASs as the modeling data set [26]. The QSPR model developed in this study uses more PFAS modeling datasets and fewer molecular descriptors to obtain better model statistical parameters, and the log K_oc_ prediction accuracy for PFASs is higher. For example, the QSPR model developed by Wang et al. has prediction errors of −0.38 and 0.18 for traditional PFASs (such as perfluorooctanoic acid (PFOA) and PFOS) [26], while the prediction errors of this model for PFOA and PFOS are only −0.29 and 0.07, respectively.

To sum up, compared with the models developed in previous studies (Table 3), the data set used in this study has more types and numbers of PFASs, the developed QSPR model has higher prediction accuracy, and the model has a wider application field. It covers a large number of structurally diverse PFASs, including not only traditional PFASs but also emerging PFASs. In addition, the modeling process of this study is completely based on the standard framework of the QSPR model [28], which not only establishes a simple linear relationship between the structural properties of PFASs and their log K_oc_ but also analyzes the distribution behavior of PFASs between sediment/soil and water in detail through the mechanical interpretation of the QSPR model.

## 3. Materials and Methods

### 3.1. Data Collection and Processing

The log K_oc_ values were collected from the research literature on the adsorption of PFASs in sediments and soils [17,21,48,49,50,51,52,53], including 11 PFCAs, 5 PFSAs, 1 perfluoroalkane sulfonamide (FOSAs), 1 perfluoroalkyl phosphinic acid (PFPiAs), and 4 other PFASs. Since the experimental data in the previous studies were measured by different experimenters and under different experimental environments, in order to ensure the reliability of the data, this study first removed the outliers that obviously deviated from the overall data samples from the multiple log K_oc_ experimental values of the same PFASs collected and then calculated the average value to develop the QSPR model.

The log K_oc_ values of 22 PFASs ranged from 1.54 to 5.04, the span range was 3.50, the average value (mean) was 3.22, and the corresponding standard deviation (SD) was 1.11. All data fell within the interval of (mean −3SD, mean +3SD) and did not require further processing [54]. In total, 80% of the data in the dataset were randomly selected as the training set (18 PFASs) for developing the QSPR model; the remaining 20% of the data were the test set (4 PFASs) for external validation of the model. Details about PFASs, experimental values and references pertaining to the modelling and external validation sets are given in Appendix A.

### 3.2. Calculation of Molecular Descriptors

The B3LYP/6-31G* algorithm in the Gaussian program package (ver. G09W, Michael F, Wallingford, CT, USA) was used to optimize the molecular structure of the PFASs in the neutral electron ground state, and the stable molecular configuration of the PFASs with the lowest energy was obtained. The Multiwfn program (ver. 3.8, Lu T, Beijing, China) [55] calculated the optimized molecular structure of the PFASs and obtained 62 molecular descriptors, including the molecular structure features, orbital energy levels, electronegativity, atomic charge, polarity, and other physical and chemical information about the PFASs. The multiple physicochemical properties of the PFASs were successfully predicted using these molecular descriptors [1,56].

### 3.3. Model Development and Validation

Firstly, correlation analysis was performed between all molecular descriptors. For molecular descriptors with a correlation coefficient (*R*) higher than 0.9, only one molecular descriptor with a high correlation coefficient with log K_oc_ was retained. Secondly, based on SPSS 26 software (IBM SPSS Inc., Chicago, IL, USA), the retained molecular descriptors were taken as independent variables with log K_oc_ as the dependent variable to perform a stepwise linear regression to obtain the QSPR models containing different numbers of molecular descriptors. Lastly, the optimal QSPR model with the largest adjusted coefficient of determination (*R*^2^_adj_) and the smallest root mean square error (*RMSE*) was selected as the final model in this study. *R*, *R*^2^_adj_ and *RMSE* were obtained by SPSS 26 software (IBM SPSS Inc., Chicago, IL, USA).

According to the QSPR model construction guidelines [28], the QSPR model should have sufficient goodness of fit, robustness, and predictive ability. In this study, the *R*^2^ was used to evaluate the goodness of fit of the QSPR model, the *Q*^2^_LOO_ was used to evaluate the robustness of the QSPR model, the test set was used to externally validate the QSPR model, and the *Q*^2^_F1_, *Q*^2^_F2_, *Q*^2^_F3_ were used to evaluate the prediction ability of the QSPR model. In addition, in order to further verify the reliability of the developed model, the error distribution of the model prediction was used to evaluate whether the model had systematic errors; the *p* and *VIF* of the molecular descriptors contained in the QSPR model were used to determine whether each molecular descriptor was significant and whether there was multicollinearity among the molecular descriptors. 

### 3.4. Application Domain

A Williams diagram [57] was used to characterize the application domain of the QSPR model, evaluate its scope of application, and determine whether there were outliers in the modeling samples. The composition of the Williams diagram and its calculation method are described in the Appendix A.

## 4. Conclusions

In this study, we successfully developed an optimal QSPR model to predict the log K_oc_ of PFASs. The dataset of this model includes 22 PFASs in eight different categories, covering the common PFASs in current industrial production and daily life, and the model has a wide range of applicability. The comprehensive verification and evaluation of the model show that the developed model has sufficient goodness of fit, robustness, and predictive ability and can accurately predict the log K_oc_ of PFASs (within the application field defined by the model).

Through the mechanistic interpretation of the model, we found that the *MW*, *E*_LUMO_, *ALIE*_max,_ and *μ* of PFASs are the main structural factors affecting the partitioning behavior between the solid and liquid phases, and the order of influence is *MW* > *E*_LUMO_ > *ALIE*_max_ > *μ*. Specifically, *MW* reflects the hydrophobic property of the compound, *μ* reflects the polarity of the compound, while *E*_LUMO_ and *ALIE*_max_ are related to the electrostatic interaction between molecules. The partitioning behavior of PFASs between the two phases is the result of the joint influence of multiple mechanisms. The hydrophobic interaction, electrostatic interaction, and dipole–dipole interaction play key roles in determining the partitioning of PFASs between the two phases. PFASs that can produce a strong hydrophobic interaction tend to be distributed in solid–phase media. The results of this study are of great significance to understanding the migration behavior and environmental fate of PFASs between sediment/soil and water, providing basic data for further environmental risk assessment.

In future research, the relationship between the log K_oc_ of PFASs and other partition coefficients can be compared and analyzed in order to illustrate the transport of PFASs in multiphase media as well. Moreover, new PFASs with less impact on the environment can be designed based on the structural factors that affect the distribution behavior of PFASs to reduce the environmental load caused by this kind of compound.

## Figures and Tables

**Figure 1 molecules-27-05610-f001:**
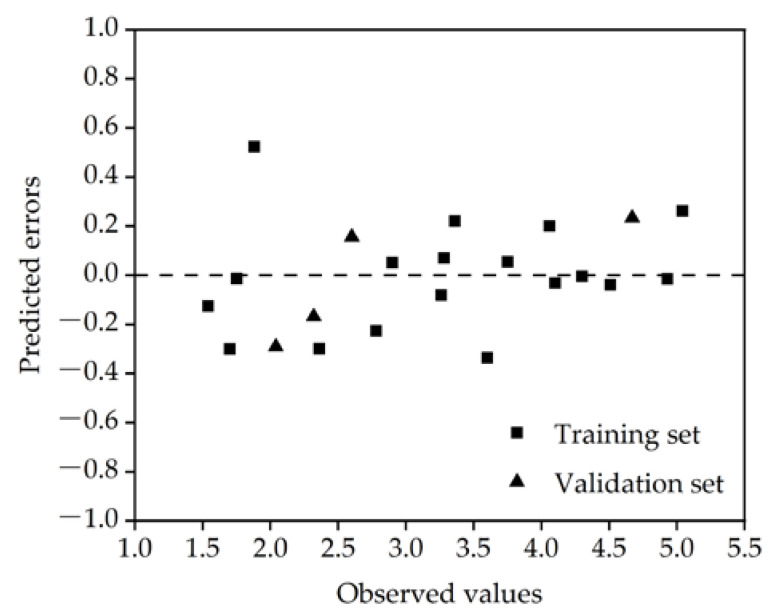
Residual diagram of the optimal model.

**Figure 2 molecules-27-05610-f002:**
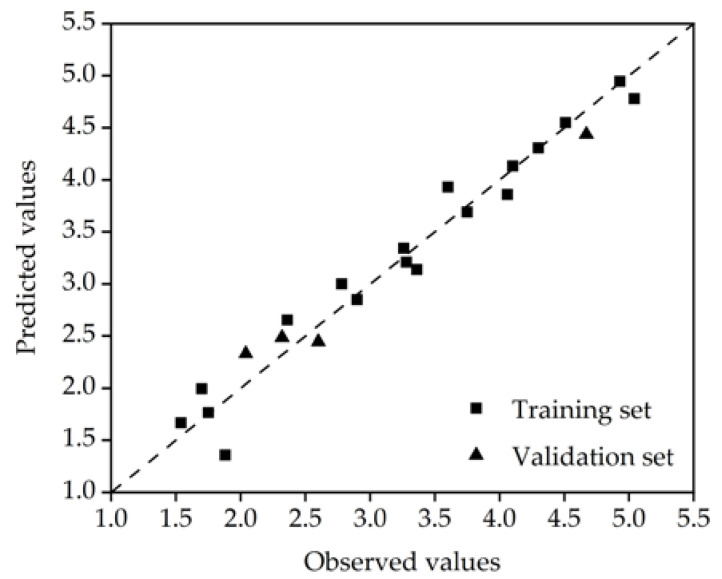
Observed and predicted values of the optimal model.

**Figure 3 molecules-27-05610-f003:**
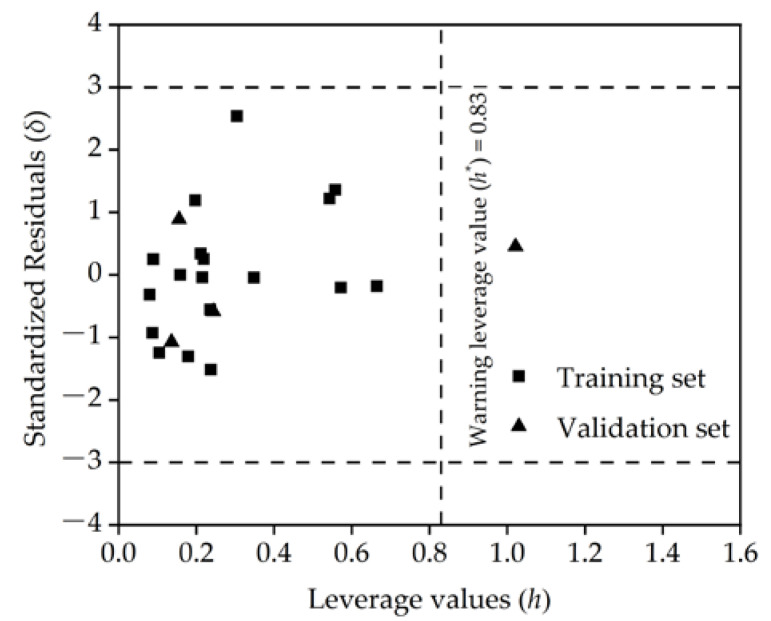
Williams diagram of the optimal model.

**Table 1 molecules-27-05610-t001:** Statistical parameters of the optimal QSPR model.

Training Set						Validation Set				
*n*	*R* ^2^	*Q* ^2^ _LOO_	*RMSE*	*F*	*p*	*n*	*Q* ^2^ _F1_	*Q* ^2^ _F2_	*Q* ^2^ _F3_	*RMSE*
18	0.962	0.920	0.212	82.269	<0.001	4	0.961	0.955	0.959	0.219

Notes: *n*: the number of data points; *R*^2^: coefficient of determination; *Q*^2^_LOO_: multiple correlation coefficient of leave-one-out cross-validation; *RMSE*: root mean square error; *F*: variance ratio; *p*: significance index; when *p* < 0.05, this indicates that the model is significant; *Q*^2^_F1_, *Q*^2^_F2_, and *Q*^2^_F3_: external validation indicators.

**Table 2 molecules-27-05610-t002:** Statistical parameters of different descriptors.

Description	*p*	*VIF*
*MW*	0.000	1.192
*μ*	0.002	1.239
*E* _LUMO_	0.000	3.420
*ALIE* _max_	0.003	3.216

Notes: *VIF*: variance inflation coefficient.

**Table 3 molecules-27-05610-t003:** Comparisons of models in the current and earlier studies.

*n*	Chemicals	*R* ^2^	*RMSE*	*Q* ^2^ _LOO_	*Q* ^2^ _F1_	*Q* ^2^ _F2_	*Q* ^2^ _F3_	Reference
12	PFCAs, PFSAs	0.98	0.200	NR	NR	NR	NR	[27]
824 *	PFCAs, PFSAs	0.854	0.472	0.850	0.761	NR	NR	[26]
22	PFCAs, PFSAs, FOSAs, PFPiAs, and other emerging PFASs	0.962	0.212	0.920	0.961	0.955	0.959	This study

Notes: NR: not reported; *: the 824 compounds in the dataset contain only six PFASs; FOSAs: perfluoroalkane sulfonamide; PFPiAs: perfluoroalkyl phosphinic acid.

## Data Availability

Further details about the data presented in this study are available on request from the corresponding authors.

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
