# Peer review of "Developing a QSPR Model of Organic Carbon Normalized Sorption Coefficients of Perfluorinated and Polyfluoroalkyl Substances"

_molecules, 2022, doi:10.3390/molecules27175610_

Round 1

Reviewer 1 Report

This study established a QSPR model to predict the PFAS distribution coefficient in soil and sediments. The novelty of this study is not high, but the methodology is generally sound. Therefore, this paper could be published after an intensive revision.

Major comments:

1. The novelty of this article is not well highlighted in the abstract and introduction. The novelty statement should be added at the end of the abstract/introduction and what you have added to the scientific society.

2. The whole Introduction should be rewritten. The hazardous effects of PFAS compounds should not be the focus. Instead, the similarities/differences between conventional PFAS and emerging PFAS in structures, and environment transmission behaviors are highly relevant to the QSPR model.

3. The research gap should be explicitly stated and linked to the objective of this study.

4. Molecular weight (MW), dipole moment (μ), lowest unoccupied molecular orbital energy (ELUMO), and maximum average local ionization energy on the molecular surface (ALIEmax) were identified as key structural variables in the model. However, the discussion on the dipole–dipole interactions impact is still weak. More evidence from the literature is required. Otherwise, it should be removed.

5. The logic in terms of how you relate model parameters to hydrophobic/electrostatic interactions is missing, especially in the abstract and conclusion sections.

6. The model performance should be evaluated based on repetitions by dividing the training set and validation set.

7. Please explicitly explain what molecular characteristics (other than MW) influence the ranking of log Koc (PFPiA > PFDoDA >EtFOSAA). The collinearity between MW with hydrophobicity and/or other properties among PFAS is interesting to discuss with more texts and insights.

Minor comments:

1. Line numbers are missing in the whole document!

2. Be more transparent in the dataset. Please include the literature citations in the table of SI.

3. Models established using advanced machine learning techniques should be also included. The results could be compared, and the model interpretability might be an advantage of this study.

4. It is highly recommended to list the 5 OECD guidelines (in SI) you mentioned in the M&M.

5. Please list all 62 parameters generated from Multiwfn and explain (using data!) how the others fail in the selection of parameter set in the final collection.

6. The unit of molar volume is missing in the context of 3.4.

Reviewer 2 Report

This study reported a QSAR study by correlating the log Koc values of different kinds of PFASs and their molecular parameters. The molecular mechanisms were proposed and this study fits well into the scope of the journal. I would suggest some minor revisions before acceptance of this paper.

(1) In the Introduction section, the authors should add more descriptions of the environmental concentrations of PFASs in different environmental media and their detailed risks.

(2) In the last paragraph of the Introduction section, the authors mentioned five guidelines for QSPR construction. I would suggest adding these five guidelines in more detail.

(3) Section 2.2: The authors should add more information on the DFT calculation, why use this calculation method? what is the solvent model?

(4) The environmental significance and implications should be highlighted in both the Abstract and Conclusion sections. 

Round 2

Reviewer 1 Report

I am generally satisfied with the answer from the authors. However, extensive English editing is still required to further polish the grammar/fluency.